# A New Strategy to Preserve and Assess Oxygen Consumption in Murine Tissues

**DOI:** 10.3390/ijms23010109

**Published:** 2021-12-22

**Authors:** Jerome Kluza, Victoriane Peugnet, Blanche Daunou, William Laine, Gwenola Kervoaze, Gaëlle Rémy, Anne Loyens, Patrice Maboudou, Quentin Fovez, Corinne Grangette, Isabelle Wolowczuk, Philippe Gosset, Guillaume Garçon, Philippe Marchetti, Florence Pinet, Muriel Pichavant, Emilie Dubois-Deruy

**Affiliations:** 1Univ. Lille, CNRS, Inserm, CHU Lille, Institut pour la Recherche sur le Cancer de Lille, UMR9020—UMR-S 1277—Canther—Cancer Heterogeneity, Plasticity and Resistance to Therapies, 59000 Lille, France; william.laine@univ-lille.fr (W.L.); anne.loyens@inserm.fr (A.L.); quentin.fovez@inserm.fr (Q.F.); philippe.marchetti@inserm.fr (P.M.); 2Univ. Lille, Inserm, CHU Lille, Institut Pasteur de Lille, U1167—RID-AGE—Facteurs de Risque et Déterminants Moléculaires des Maladies Liées au Vieillissement, 59000 Lille, France; victoriane.peugnet@univ-lille.fr (V.P.); florence.pinet@pasteur-lille.fr (F.P.); emilie.deruy@pasteur-lille.fr (E.D.-D.); 3Univ. Lille, CNRS, Inserm, CHU Lille, Institut Pasteur de Lille, U1019—UMR9017—CIIL—Center for Infection and Immunity of Lille, 59000 Lille, France; blanche.daunou@univ-lille.fr (B.D.); gwenola.kervoaze@pasteur-lille.fr (G.K.); remy.gaelle@aol.fr (G.R.); corinne.grangette@pasteur-lille.fr (C.G.); isabelle.wolowczuk@pasteur-lille.fr (I.W.); philippe.gosset@pasteur-lille.fr (P.G.); 4Centre de Bio-Pathologie, CHRU Lille, 59000 Lille, France; patrice.maboudou@chru-lille.fr; 5Univ. Lille, CHU Lille, Institut Pasteur de Lille, ULR 4483-IMPacts de l’Environnement Chimique sur la Santé (IMPECS), 59000 Lille, France; guillaume.garcon@univ-lille.fr

**Keywords:** oximetry, oxidative phosphorylation, energy metabolism, high fat diet

## Abstract

Mitochondrial dysfunctions are implicated in several pathologies, such as metabolic, cardiovascular, respiratory, and neurological diseases, as well as in cancer and aging. These metabolic alterations are usually assessed in human or murine samples by mitochondrial respiratory chain enzymatic assays, by measuring the oxygen consumption of intact mitochondria isolated from tissues, or from cells obtained after physical or enzymatic disruption of the tissues. However, these methodologies do not maintain tissue multicellular organization and cell-cell interactions, known to influence mitochondrial metabolism. Here, we develop an optimal model to measure mitochondrial oxygen consumption in heart and lung tissue samples using the XF24 Extracellular Flux Analyzer (Seahorse) and discuss the advantages and limitations of this technological approach. Our results demonstrate that tissue organization, as well as mitochondrial ultrastructure and respiratory function, are preserved in heart and lung tissues freshly processed or after overnight conservation at 4 °C. Using this method, we confirmed the repeatedly reported obesity-associated mitochondrial dysfunction in the heart and extended it to the lungs. We set up and validated a new strategy to optimally assess mitochondrial function in murine tissues. As such, this method is of great potential interest for monitoring mitochondrial function in cohort samples.

## 1. Introduction

Mitochondrial metabolism and glycolysis play critical roles in a variety of cellular processes, including cellular activation, proliferation, differentiation, and death, as well as, more broadly, disease progression [1]. Mitochondrial function is commonly evaluated through oxygen consumption measurements using Clark electrodes [2], such as in high-resolution respirometry [3]. Another method, the XFe24/XF96 Extracellular Flux Analyzer (Seahorse Bioscience), uses oxygen-sensing fluorophores to assess mitochondrial oxygen consumption in freshly isolated mitochondria [4,5], permeabilized fibers [6], intact cells [7,8] or fresh tissues [9].

Therefore, ex vivo techniques on isolated cells and mitochondria may not adequately reflect the in vivo status, since cells and organelles will not be in their tissular and cellular micro-environments, respectively. Moreover, isolation procedures often altered cell and organelle structures and functions, notably through disrupting the mitochondrial network, due to the loss of intracellular interactions [10,11]. Most often individual cell populations are analyzed, losing the cell-to-cell interactions, which occur in the whole tissue [11]. Analysis of permeabilized cells induced a lack of most of the cytoplasm (thus the impact of cytosolic factors and intra-tissular nutrients cannot be studied) and a risk of partial or excessive permeabilization [11].

To overcome limitations of the ex vivo approaches, several in vivo techniques have been used to study mitochondrial function within the context of the physiological environment. The most commonly used methods are nuclear magnetic resonance (NMR) spectroscopy. These techniques allow to monitor mitochondrial metabolism in vivo as well as investigate how it is modulated in health and disease, however they are not implemented in every laboratory [12]. In the present study, we proposed a complementary approach to assess mitochondrial metabolism in non-permeabilized lung and heart tissue. 

With regards to lifestyle and environmental factors impact on health, it is well established that unhealthy diet consumption (such as high-fat diet (HFD) leads to obesity [7] and its related metabolic abnormalities, increasing the risk for cardiac hypertrophy [4,13] and lung chronic inflammatory disease [7]. In addition, chronic exposure to HFD can have a detrimental effect, due to the accumulation of toxic metabolic derivatives, such as reactive oxygen species (ROS), and/or mitochondrial dysfunction [5]. To investigate the mitochondrial (dys)function in whole heart and lung tissues, we set up and validated a new strategy on preserved tissues using the XFe24 Extracellular Flux Analyzer (Seahorse) technology. We evaluated the impact of overnight conservation of murine heart and lung tissues on oxygen consumption, macroscopic and histologic appearance, and mitochondrial ultrastructure, compared to fresh tissues. We validated our method using murine heart or lung tissues isolated from lean or obese mice. Altogether, we provided evidence that this procedure might be a useful tool for monitoring mitochondrial function in cohort samples.

## 2. Results

### 2.1. Quantification of Mitochondrial Respiration in Heart and Lung Tissue Samples 

We designed a new protocol for evaluating oxygen (O_2_) consumption rates (OCR) in murine heart and lung tissues, using the XFe24 Extracellular Flux Analyzer. First, multiple transversal cuts were performed with a scalpel, starting from the left ventricular apex for the heart (according to the orientation of the fibers), and from the main bronchi for the lung (Figure 1A). Size-matched tissue samples were prepared as thin as possible, yet large enough to cover the bottom of a Seahorse Islet Plate’s well. Tissue samples were maintained at the bottom of the plate, thanks to the grid provided with the Seahorse Islet Plate (Figure 1B,C). To limit variability between wells and, therefore, differences in mitochondrial metabolism readouts, samples of comparable area were selected for each tissue: ~1 mm^2^ for heart and 0.5 mm^2^ for lung (Figure 1D), which corresponded to, respectively, 200 and 80 µg of proteins (Figure 1E). Due to the variable area of the tissue samples analyzed (Figure 1D), normalization by protein amount was required in the following experiments to adequately compare the different conditions. 

We then measured O_2_ concentrations (mmHg) in the medium of each well containing cardiac left ventricles (Figure 2) and lung (Figure 3) samples. Table A1 recapitulates the parameters that were used to assess the O_2_ levels, OCR, ratios between mitochondrial vs. non-mitochondrial respiration and extracellular acidification rates (ECAR) of the different samples. Our experimental conditions allowed for maintaining a high level of O_2_ during all the procedure, therefore avoiding reaching hypoxia or anoxia conditions, which are toxic for tissues and cells. We checked that our experimental conditions permitted the total reoxygenation of the medium to allow the next measurement, knowing that the range of O_2_ concentration was stable in the vehicle condition (Figure 2A and Figure 3A). Next, we used the well-characterized mitochondrial inhibitors rotenone (rot) and antimycin A (AA), which, respectively, inhibited the complexes I and III of the mitochondrial respiratory chain. First, we realized a titration of rotenone and antimycin A to determine the appropriate concentration to reach the maximal inhibition of mitochondrial respiration (Figure A1). With both drugs, a full inhibition of the mitochondrial respiration was observed after 90 min in both heart and lung tissues (Figure 2B and Figure 3B). We noticed that the respiration rate slightly decreased after 50 min in the absence of inhibitors in cardiac left ventricles (Figure 2B). This was not observed in lung samples (Figure 3B). 

Next, the ratios between mitochondrial vs. non-mitochondrial respiration were determined in the tissue samples and the corresponding cell-lines H9c2 (for heart, Figure 2C) and A549 (for lung, Figure 3C). Rotenone and Antimycin A inhibited mitochondrial oxygen consumption in both models, but there was a more important insensitive rotenone/antimycin respiration in the tissues than in the corresponding cell lines, for the heart (Figure 2C, Figure 3C and Figure A2). Similar results have been obtained with potassium cyanide, an inhibitor of the complex IV of the mitochondrial respiratory chain (Figure A3A,B). 

This elevated non-mitochondrial respiration in cardiac tissue may be due to stress following anesthesia prior to mouse sacrifice [14].

### 2.2. Impact of the Conditions of Murine Tissue Conservation on Mitochondrial Integrity and Function

Heart (Figure 2) and lung (Figure 3) samples were kept at 4 °C or 37 °C overnight and compared to fresh tissues. Macroscopically, we did not observe any obvious changes between fresh tissues and the tissues stored at 4°C overnight. These observations were confirmed by photonic microscopy after hematoxylin/eosin staining, since no significant modifications of the heart or lung tissue histology were evidenced (Figure 2D and Figure 3D). In striking contrast, tissues kept at 37 °C overnight seemed to be altered. Indeed, cardiac biopsies began to lose cross striations and the nuclei were neither clearly visible nor centrally placed in most of the cells (Figure 2D). Similarly, lung biopsies had altered tissue histology, as evidenced by parenchymal inflammation and lesions, bronchial damages, and necrosis (Figure 3D). 

Accordingly, no significant ultrastructural changes and no mitochondrial size modification were evidenced between fresh tissues and tissues stored at 4 °C overnight (Figure 2E,F, Figure 3E,F and Figure A4). In both conditions, tissues had normal mitochondrial cristae and inner/outer mitochondrial membranes, suggesting preserved physiological and functional organization of the mitochondria. In marked contrast, heart and lung ultrastructures were entirely disorganized after overnight conservation at 37 °C, confirming that this condition was deleterious for the tissue’s integrity preservation.

These results were corroborated by the evaluation of cell membrane disruption (relative to necrosis) assessed through lactate dehydrogenase (LDH) release in biopsies supernatants: the LDH activity remained low and comparable between fresh tissues and tissues stored at 4 °C overnight, for both heart (Figure 2F) and lung (Figure 3F). Oppositely, LDH activity was significantly increased after overnight conservation at 37 °C for both tissues (*p* = 0.001 for heart, and *p* = 0.014 for lung). Preservation of tissue overnight at 4 °C did not induce hypoxia as depicted by mRNA levels of target genes regulated by HIF-1α (heme oxygenase 1, Pyruvate dehydrogenase kinase 3, lactate dehydrogenase 2, and GLUT transporter SLC2A1) in contrast to 37 °C in heart (Figure A5A) and lung (Figure A5B). Altogether, these results showed that overnight conservation of heart and lung samples at 4 °C neither altered tissue viability or organization, nor induced hypoxia.

To further decipher the impact of conservation conditions on mitochondrial respiratory functions, we used the protocol above described (Figure 2A and Figure 3A, as well as Table A1) using rotenone and antimycin A. We compared OCR in cardiac left ventricles (Figure 2G) and lung (Figure 3G) samples, either fresh or conserved overnight at 4 °C or 37 °C. We did not observe any significant change in OCR between fresh and 4 °C overnight-stored tissues. In contrast, the OCR values of tissues kept at 37 °C overnight were drastically reduced, almost reaching the background level, in accordance with the observed damaged mitochondria (Figure 2E, Figure 3E and Figure A4).

### 2.3. Validation of the Method by Monitoring the Mitochondrial Metabolism in Metabolically-Altered Tissues

As a “proof-of-concept”, we tested our method with heart and lung samples harvested from obese and lean mice. We chose this model since other teams reported that cells isolated from organs of obese mice (high-fat diet-induced obesity) exhibited a higher OCR when compared with cells isolated from lean mice [5,7]. As shown in Figure 4A,B, sixteen weeks of a high-fat diet (HFD) induced a significant increase in body weight (*p* < 0.0001) and an increase of plasma triglyceride and cholesterol concentrations when compared to low-fat diet (LFD)-fed lean animals. Heart and lung tissues were collected from lean and obese animals and stored at 4 °C overnight. OCR and extracellular acidification rate (ECAR) values were then determined in tissue samples. As expected from literature, HFD induced a significant increase in OCR and ECAR in cardiac left ventricles (*p* = 0.0046) (Figure 4C,D). Interestingly, we showed that the OCR and ECAR rates were increased in the lung tissue of obese mice (*p* = 0.0052) (Figure 4E,F), compared to lean animals. This experiment confirmed that mitochondrial metabolic behavior is impaired in heart of HFD-mice and highlighted similar findings in the lung of obese mice. Importantly, we demonstrated in pathologic situations that whole biopsies kept at 4 °C overnight can be used for measuring mitochondrial metabolism.

## 3. Discussion

To investigate the mitochondrial (dys)function in whole tissues, we set up a novel procedure for monitoring mitochondrial oxidative phosphorylation in cardiac left ventricles and lung tissue samples using the XF24 Extracellular Flux analyzer. We showed that overnight conservation of tissue samples at 4 °C did not impact the mitochondrial respiration, when compared to freshly treated samples. We further validated our approach in the pathophysiological context of obesity. This could be proposed as a potential clinical tool to monitor mitochondrial function (Figure 5).

Several methods have been described to evaluate mitochondrial functions either from isolated mitochondria or from dissociated cells from fresh tissues [15,16]. Assessment of enzymatic activities of mitochondrial respiratory chain could be easily performed from frozen tissues, [17] yet all these analyses could not adequately reflect the oxygen consumption of in vivo situation. Assessment of oxygen consumption from isolated mitochondria (extracted from fresh tissue) measured mitochondrial metabolism in an acellular environment [18]. In this context, the physiological process of fusion/fission of mitochondria was altered and the oxygen consumption measured could not reflect the situation of the whole cell, and therefore, the whole tissue (including the intra-tissue nutrients). Oxidative phosphorylation could be determined from dissociated cells, but they would suffer from the mechanical/physical/enzymatic methods used for tissue disruption. Moreover, this method altered cell shape and cytoskeleton and therefore, the multicellular organization. Assessment of mitochondrial oxygen consumption from tissue sample described in this study was another complementary approach. Using this procedure, cells were kept closely to their physiological environment: tissue architecture and cell-cell interactions were preserved, therefore reflecting the metabolic activity of the whole organ [16,17]. As an example, if OCR was decreased, it could suggest that the metabolic activity of the tissue was altered and that the associated organ could not be fully functional. However, the defective cell type responsible for the observed dysfunction could not be identified. Moreover, the tissue outcome could not necessarily reflect the defective status of a minor cell type. Thus, the reliable and easily set up technique that we described herein could be used as a preliminary screening of tissue metabolic disorders.

Interestingly, our study demonstrated that heart and lung samples which have been stored overnight at 4 °C (non-hypoxic conditions) exhibited an unmodified respiration when compared to fresh tissue obtained shortly after mouse sacrifice. We showed that our new reliable and easily set up method allowed to demonstrate that HFD induced an increase in OCR consumption in cardiac and lung tissue sections. This metabolic behavior has already been observed in cells isolated by collagenase disruption from kidney [5] and lung [7] of obese mice in comparison with cells isolated from lean mice. Moreover, preserving samples overnight at 4 °C could be useful to collect many samples from large cohorts and to compare their OCR on the same plate, limiting experimental errors as we compared mitochondrial metabolism of heart and lungs of obese/lean mice. In this pathological context, we failed to evaluate the OCR from various adipose tissues (visceral, subcutaneous, and brown) due to the large heterogeneity of the samples and the low cell content.

However, this method has some limitations. The heart is a contracting muscle, which creates a permanent energetic demand for mitochondria. This method measures respiration in non-contracting heart tissue and probably underestimates oxygen consumption of the tissue. This problem could be probably solved by permeabilization of the tissue with digitonin or saponin, as well as the addition of ADP to mimic a high-energy demand, but this methodology leads to intracellular component escape. Exposure to pronophonore molecules such as carbonyl cyanide-4-(trifluoromethoxy)phenylhydrazone (FCCP) or (2-fluorophenyl){6-[(2-fluorophenyl)amino](1,2,5-oxadiazolo[3,4-e]pyrazin-5-yl)}amine (Bam15) could be another alternative to determine the maximal oxygen consumption rate of cardiac tissue. Moreover, a protocol was recently described to prepare highly viable adult ventricular myocardial slices [19]. If these contractile slices are compatible with Seahorse technology, this could further increase the interest of our method described here.

Culture medium used during the assay was based on usual formulations described by many other studies [20,21], but this standardized formulation was characterized by supraphysiological concentrations of substrates. Moreover, the addition of some metabolites could be more appropriate depending on the tissue origin. It is well established that cardiomyocytes use preferentially fatty acids as metabolites [22]. In our experimental settings, the oxygen consumption, and therefore the metabolic profile of heart tissue, could have been biased by the medium containing glucose and glutamine, favoring glycolysis.

Pathological and physiological evidence revealed mitochondrial alterations in numerous major diseases. Mitochondrial oxygen consumption is now considered as a potential biomarker. In heart disease, mitochondrial metabolism is involved in coronary atherosclerosis, ischemia/reperfusion injury, hypertension, obesity, metabolic syndrome, and diabetes, as well as cardiac hypertrophy [23,24]. In lung diseases, mitochondrial activity has been implicated in the development of chronic obstructive pulmonary disease, idiopathic pulmonary fibrosis, asthma, and lung cancer [25]. Here, we have described and validated a methodology to evaluate the mitochondrial oxygen consumption in whole tissue, which could be useful to evaluate the pharmacological approaches protecting or restoring mitochondrial activity in the pathological mouse models.

In many experiments, mitochondrial activity is a marker of cell viability. Mitochondrial activity can be easily assessed in monolayer cells by MTT assay (based on enzymatic activity of mitochondrial succinate) or by chemiluminescent measurement of ATP production. Yet these methods cannot easily be used to assess cell viability in the tissue context. As seen in this work, morphological analysis was widely used to determine cell viability, but this approach was rather qualitative than quantitative. The evaluation of mitochondrial oxygen by XFe Seahorse, as described here, can easily quantify tissue integrity. As a new method that could be used rapidly to screen metabolic activity of tissues/organs, we could translate our approach to clinic. In the critical field of heart and lung transplantation, the survival rate at 5-years is only of 65% and 40%, respectively. These organs are particularly sensitive to environmental factors, including donor surgery, transport and conservation conditions (6 h maximum), and recipient surgery. Moreover, dysfunction of mitochondrial metabolism contributes to cell damage and poor transplant results [26].

Until now, organ quality was only macroscopically verified by a member of the transplant team before transplantation, without evaluating the metabolic status of these organs. Since mitochondrial metabolism is essential to cell survival, our rapidly and easily set up procedure, using small biopsies and feasible within 2 h, could be used to confirm the quality of heart or lung transplant before organ transplantation surgery.

## 4. Materials and Methods

### 4.1. Chemicals

Rotenone (R8875) and antimycin A (A8674), were purchased from Sigma-Aldrich (St Louis, MO, USA). Dulbecco/Vogt modified Eagle’s minimal essential medium (DMEM) (D5030, Gibco), glutamine (25030149, Gibco), and glucose (A2494001, Gibco) were purchased from ThermoFisher Scientific (Carlsbad, CA, USA).

### 4.2. Murine Samples

Eight-week-old male C57BL/6J mice were purchased from Janvier (Le Genest-St-Isle, France). Animals were housed in specific pathogen-free conditions in the animal facilities of the Institut Pasteur de Lille (accredited no. A59107) and maintained in a temperature-controlled (20 ± 2 °C) facility with a strict 12-h dark/light cycle. Before experimentation, animals were provided a one-week acclimation period and were given ad libitum access to food and water. At the end of the protocol, mice were euthanized using cervical dislocation according to our Guidelines.

Sample preparations

The left lobe of the lungs and the whole heart were collected from C57/BL6J male mice and put onto ice. Tissues were then freshly processed or kept in 1 mL of Roswell Park Memorial Institute (RPMI) medium completed with 10% fetal bovine serum (FBS), either at 4 °C or at 37 °C overnight until use. Transversal cuts (~1 mm^3^ for heart and ~0.5 mm^3^ for lung) were performed with a scalpel starting from the main bronchi for the lung sample, and the apex of the left ventricle for the heart, as indicated in Figure 1A.

Experimental obesity model

High-fat diet (60% kcal fat; D12492) and low-fat diet (10% kcal fat; D12450B) were both purchased irradiated from Research Diets (Brogaarden, Lynge, Denmark). Mice were randomly assigned to be fed either with LFD diet (*n* = 20) or HFD diet (*n* = 26) for 16 weeks as we previously described [27].

### 4.3. Light Microscopy

Cardiac left ventricle and lung samples were fixed in formalin and provided to Oncovet Clinical Research (Loos, France) for histology analysis. After paraffin embedding, sections (4 µm) were then stained with Hematoxylin-Eosin-Saffron. Histopathological assessments were blindly performed by a veterinary pathologist.

### 4.4. Lactate Dehydrogenase Activity Assay

Supernatants of heart and lung tissues incubated in the different conditions of conservation were collected and lactate levels were measured on a SYNCHRON LX20 Clinical system (Beckman Coulter, Fullerton, CA, USA).

### 4.5. Oximetry Assessment

The XFe24 Extracellular Flux Analyzer (Seahorse XF, Seahorse Bioscience Inc.) is a 24-well instrument that measures in real-time oxygen consumption rate (OCR) and extracellular acidification rate (ECAR) relative to glycolysis activity. Sensor cartridges, containing fluorescence probe, were hydrated with XF Calibrant (Agilent Technologies) the day before the assays, according to manufacturer’s instructions, and preloaded with the freshly prepared inhibitors rotenone and antimycin A, 30 min before running the assay. Prior to start the XF assay, biosensors were individually calibrated using the standard automated routine. A low-buffered DMEM medium (assay medium) containing glucose (10 mM) and glutamine (2 mM) was extemporarily prepared and adjusted to pH 7.4 before the assay. Fresh slices of heart (~1 mm^2^) and lung (~0.5 mm^2^) tissues were laid over the XFe 24-well islet capture microplate wells (Seahorse Bioscience) in 200 µL of 37 °C pre-warmed assay medium and then incubated at 37 °C without CO_2_ for 30 min. Then, 300 μL of pre-warmed (37 °C) assay medium were added and equilibration performed after calibration. Prior to each rate measurement, mixing of the media in each well was performed by the XFe24 Analyzer to allow the oxygen pressure to reach equilibrium. OCR and ECAR were then measured simultaneously (see Table A1 for parameter settings). After the baseline measurement (at least 4 independent measures), 75 μL of rotenone/antimycin A mixture was injected into each well to reach a final concentration of 10 μM for both inhibitors and followed by mixing (see Table A1 for parameters). At the end of each assay, tissues were removed from the well for protein quantification assay.

### 4.6. Protein Quantification Assay

At the end of Seahorse experiments, samples were lyzed through Dounce-Potter homogenization into 100 µL of ice-cold RIPA buffer (Sigma-Aldrich) and kept at −20 °C until analysis. The protein quantification assay was performed with Pierce™BCA Protein Assay Kit (ThermoFisher) following manufacturer’s instructions.

### 4.7. Electron Microscopy

Samples were fixed with paraformaldehyde 2%/picric acid 0.2%/glutaraldehyde 1% at 4 °C for 72 h, and then washed with phosphate buffer 0.1 M pH 7.5. Samples were then post-fixed in 1% osmium tetroxide in phosphate buffer 0.1 M at room temperature (RT) for 1 h then washed with phosphate buffer 0.1 M pH 7.5; followed by dehydration steps (5 min in ethanol 50%, 5 min in ethanol 70%, 5 min in ethanol 80%, 2 × 15 min in ethanol 95%, 3 × 20 min in ethanol 100%, and 20 min in propylene oxide). Samples were then stained in propylene oxide/Araldite^®^ (*v/v*) for 1 h, then in Araldite^®^ 100% at 4 °C overnight before capsules embedding at 56 °C overnight.

Ultrathin sections (85 nm) were performed with a Leica UM EC7 ultramicrotome, and contrast of sections were performed by uranyle acetate 2%/ethanol 50% treatment for 8 min followed by Reynolds lead citrate for 8 min. Sections were observed using a Zeiss EM900 electron microscope.

### 4.8. RNA Extraction and Quantitative RT-PCR

Total RNA from cardiac left ventricles and lungs was prepared using TRI Reagent (Sigma-Aldrich) according to the manufacturer’s instructions. RNA (500 ng) from each sample was reverse transcribed with miScript II RT Kit (Qiagen). qPCR was then performed using the miScript SYBR Green PCR Kit (Qiagen). All samples were processed in duplicate reactions on a Stratagene Mx3005O (Agilent Technologies). Primer sequences used were Heme Oxygenase 1 (HO-1) (forward: 5′-AAGCCGAGAATGCTGAGTTCA-3′; reverse: 5′-GCCGTGTAGATATGGTACAAGGA-3′) and Hypoxanthine guanine phosphoribosyltransferase (HPRT) (Forward: AAAGGACCTCTCGAAGTGTT; Reverse: TGACACAAACGTGATTCAAA). Relative mRNA levels (2-Δ(ΔCt)) were determined by comparing: (1) the PCR cycle thresholds (Ct) for the gene of interest and the housekeeping and (2) ΔCt values for the different conditions (ΔΔCt).

### 4.9. Cell Cultures

The rat embryonic heart-derived H9c2 cell-line (ATCC, CRL-1446) and the human epithelial lung carcinoma cell-line (ATCC, CCL-185) were cultured following the manufacturer’s instructions.

### 4.10. Statistics

Statistics were performed with GraphPad Prism^®^ version 6.0 (GraphPad Software, San Diego, CA, USA). Data are presented as the mean ± S.E.M. Differences between measurements in groups were determined by Kruskall-Wallis with Dunn’s post-test analysis when sample size < 6, and Student’s *t*-test when sample size > 6. *p* < 0.05 was considered significant with *p* < 0.05 indicated as (*), *p* < 0.01 as (**), and *p* < 0.001 as (***).

## 5. Conclusions

In the end, we set up and validated a new strategy to optimally assess mitochondrial function in murine tissues. As such, this method is of great potential interest for monitoring mitochondrial function in cohort samples.

## Figures and Tables

**Figure 1 ijms-23-00109-f001:**
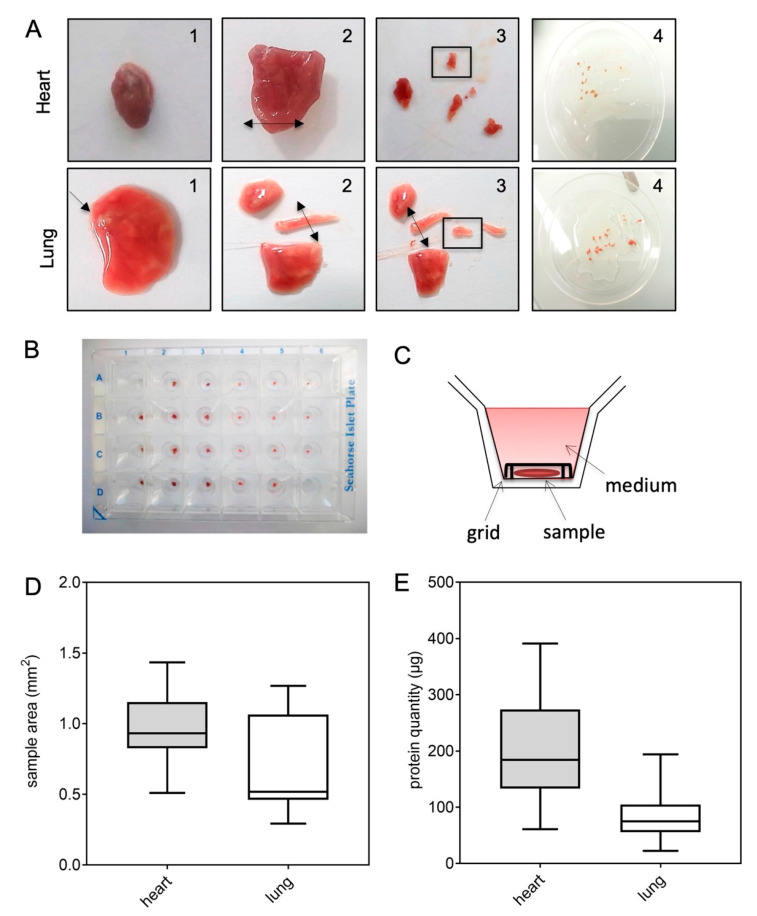
Sample preparation for reproducible respirometry experiments. Samples were obtained from heart and lung tissues and processed as shown in (**A**). Transversal cuts were performed with a scalpel, starting from the left ventricular apex for the heart, and the main bronchi for the lung, as indicated by the arrows (1 and 2). Size-matched samples (framed) were prepared as thin as possible (3 and 4). One sample per well was then placed in a Seahorse Islet plate (**B**). Each sample was maintained at the bottom of the plate using the grid provided with the Seahorse Islet plate before filling the well with medium (**C**). Samples mean area (mm^2^) (**D**), and protein quantity (µg) (**E**) were obtained for the optimal quantification of the oxygen consumption rates (OCR) in heart (grey) and lung (white) tissues (*n* = 10).

**Figure 2 ijms-23-00109-f002:**
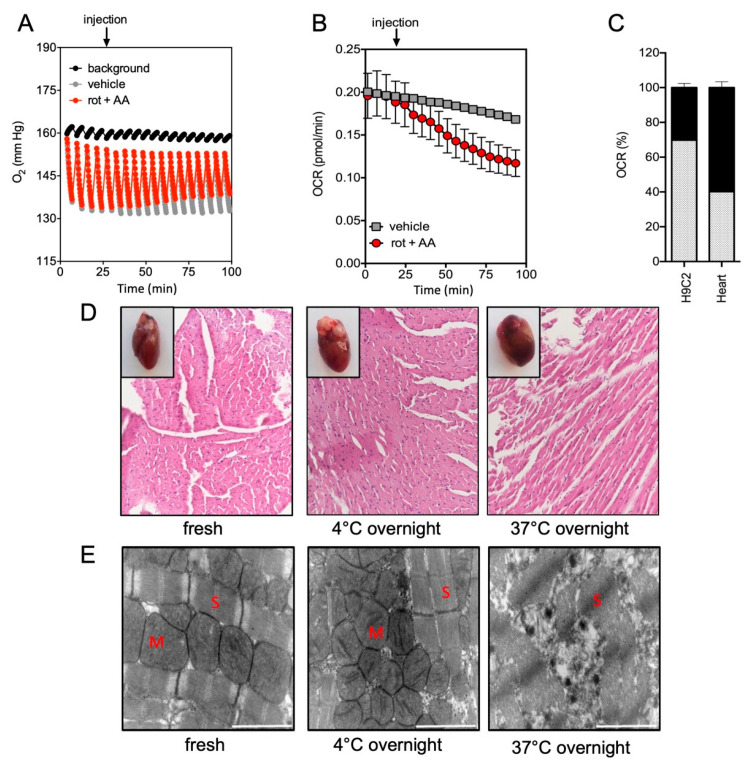
Oxygen (O_2_) consumption measurements and the effect of conservation conditions in cardiac left ventricle samples. O_2_ concentration (mmHg) (**A**), oxygen consumption rate (OCR, pmol/min/µg protein) (**B**) and mitochondrial OCR (%) (**C**) were quantified in cardiac left ventricle samples. In **A**, background (black line) corresponded to empty wells. A mix of rotenone (rot) and antimycin A (AA) was injected at the indicated time (arrow) in order to progressively inhibit mitochondrial respiration (red) and was compared to vehicle injection (grey). OCR had been assessed from duplicate biopsies of three mice for each condition (*n* = 6) (**B**). Mitochondrial O_2_ consumption was calculated from basal OCR (obtained without inhibitor, grey) subtracted to OCR level after rot/AA treatment (non-mitochondrial respiration, black). Both values were compared as a percentage of basal OCR for tissue samples and the cardiac cell line H9c2 (**C**). Representative histological sections (Hematoxylin-Eosin-Saffron staining) of fresh and conserved overnight at 4 °C and 37 °C heart samples (**D**). The corresponding macroscopically views were inserted in each panel. (**E**) Representative electronic microscopy images of heart (×20,000 magnification, scale bar: 2 µm) (S: Sarcomeres, M: Mitochondria); (**F**) Quantification of mitochondrial size (cross-sectional area). Approximately 100 mitochondria from four different hearts or lungs were analyzed. (**G**) Lactate dehydrogenase (LDH) activity (U.L-1) was quantified in the medium surrounding the heart (*n* = 3) as a surrogate of potential tissue damage. Graphs showed the individual and mean values. Significant *p* values are indicated, ** *p* < 0.001. (**H**) Quantification of OCR (pmol/min/µg protein) in cardiac left ventricles samples stored in the different conditions: fresh (white), 4 °C overnight (grey) and 37 °C overnight (black). OCR had been assessed from duplicate biopsies of three mice for each condition (*n* = 6). Injection of rotenone (rot) and antimycin A (AA) was indicated by an arrow.

**Figure 3 ijms-23-00109-f003:**
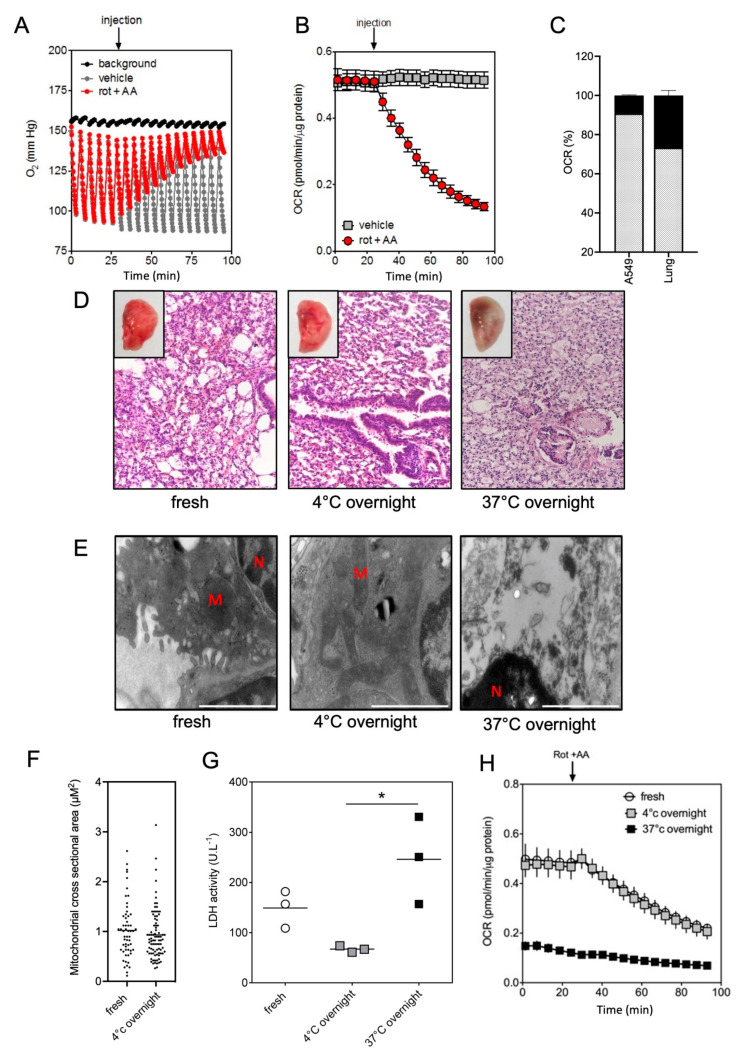
Oxygen (O_2_) consumption measurements and the effect of conservation conditions in lung samples. O_2_ concentration (mmHg) (**A**), oxygen consumption rate (OCR, pmol/min/µg protein) (**B**) and mitochondrial OCR (%) (**C**) were quantified in lung samples. In (**A)**, background (black line) corresponded to empty wells. A mix of rotenone (rot) and antimycin A (AA) was injected at the indicated time (arrow) in order to progressively inhibit mitochondrial respiration (red) and was compared to vehicle injection (grey). OCR had been assessed from duplicate biopsies of three mice for each condition (*n* = 6) (**B**). Mitochondrial O_2_ consumption was calculated from basal OCR (obtained without inhibitor, grey) subtracted to OCR level after rot/AA treatment (non-mitochondrial respiration, black). Values were compared as a percentage of basal OCR for tissue samples and the lung cell line A549 (*n* = 3) (**D**). Representative histological sections (Hematoxylin-Eosin-Saffron staining) of fresh and conserved overnight lung samples at 4 °C and 37 °C (**D**). The corresponding macroscopical views were inserted in each panel. (**E**) Representative electronic microscopy images of lung (×30,000 magnification, scale bar: 1 µm); (N: Nucleus, M: Mitochondria); (**F**) Quantification of mitochondrial size (cross-sectional area). Approximately 100 mitochondria from four different hearts or lungs were analyzed. (**G**) Lactate dehydrogenase (LDH) activity (U.L-1) was quantified in the medium surrounding the lung (*n* = 3) as a surrogate of potential tissue damage. Graphs showed the individual and mean values. Significant *p* values are indicated, * *p* < 0.05. (**H**) Quantification of OCR (pmol/min/µg protein) in lung samples stored in the different conditions: fresh (white), 4 °C overnight (grey) and 37 °C overnight (black). OCR had been assessed from duplicate biopsies of three mice for each condition. Injection of rotenone (rot) and antimycin A (AA) was indicated by an arrow.

**Figure 4 ijms-23-00109-f004:**
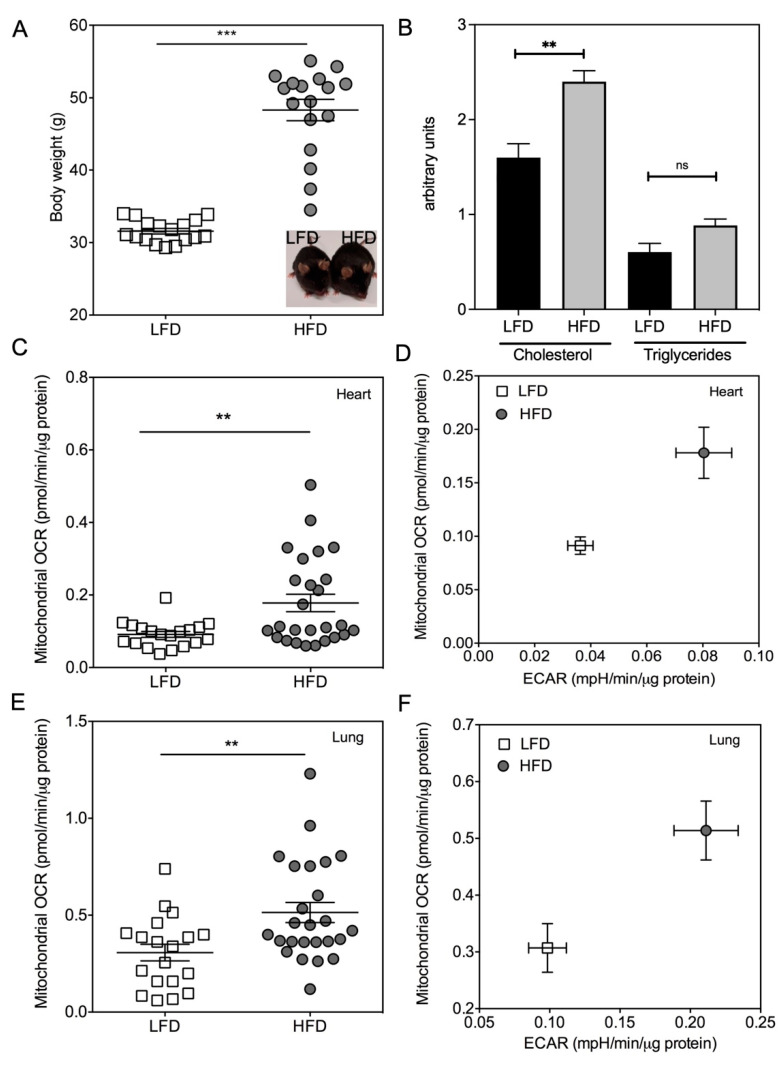
Impact of high-fat diet (HFD) feeding on oxygen consumption in cardiac and lung samples. (**A**) Sixteen weeks of high-fat diet (HFD) induced a significant increase in body weight compared to low-fat diet (LFD). (**B**) Quantification of cholesterol and triglycerides in plasma of LFD and HFD mice. (**C**,**E**) Mitochondrial oxygen consumption rate (basal OCR—non mitochondrial OCR) and (**D**,**F**) extracellular acidification rate (basal ECAR) have been assessed from biopsies of at least 20 mice for each diet in heart and lung samples. Graphs showed the individual and mean values. Significant *p* values are indicated, ** *p* < 0.01. *** *p* < 0.001.

**Figure 5 ijms-23-00109-f005:**
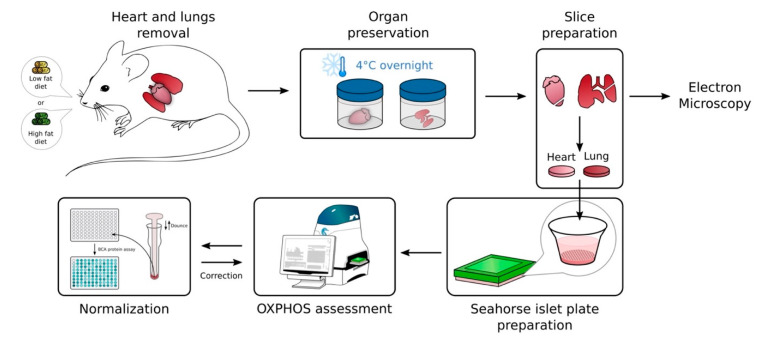
Schematic representation of the methodology used to measure oxygen consumption in 4 °C conserved tissues with XFe24 Seahorse. Heart and lung samples have been freshly removed from untreated/treated mice and quickly stored at 4 °C overnight. Next day, tissues have been dissected and cut in small pieces of tissue for electron microscopy and oxygen consumption (OXPHOS assessment) analysis. Following measurements, protein concentrations of tissues have been determined to normalize oxygen consumption rate.

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
