# Peer review of "A New Strategy to Preserve and Assess Oxygen Consumption in Murine Tissues"

_ijms, 2021, doi:10.3390/ijms23010109_

Round 1

Reviewer 1 Report

The study from Kluza et al is interesting and addresses an important topic i.e. measurement of tissue respiratory capacity. The authors describe a new method of assessing O2 consumption in intact tissues and addressed the viability of a ‘’storing’’ option in this methodology (storing samples overnight at 4 degrees). In addition, the authors assessed O2 consumption rate in cardiac and lung tissue of obese mice. Although the methodology and results that are being described are interesting, I do have a few concerns/remarks.

Major comments

  1. Statements regarding (abnormal) histology and mitochondrial ultrastructure without proper quantification of specific features warrants caution and this should be addressed. In this context, depiction of histological or ultrastructural features (e.g. Fig. 2 D, E; Fig 3D, E; Fig S2C) without further labelling of specific intracellular features (organelles) is unclear.

  1. For the HFD vs control diet experiment using mice, assessment of specific blood circulatory markers (at least cardiometabolic markers such as triglycerides, cholesterol etc) including substrates for mitochondrial metabolism would strengthen the manuscript. In addition, OCR measurements using specific substrates (for example fatty acids e.g. palmitoyl carnithine) could shed more light on mitochondrial substrate use in the obese animals vs control animals.

  1. The rationale and impact for assessing OCR in lung tissue in the context of obesity should be described more clear in the introduction and should be more elaborately discussed in the discussion.

  1. Assessment of HO1 mRNA as a marker for hypoxia in the tissues should be expanded to more hypoxia-related markers. For example assessment of HIF-1a protein levels or measurements of more HIF-responsive genes.

  1. The extent to which assessment of OCR in peripheral lung tissue vs BEAS-2B cells is a viable comparison can be debated as peripheral lung tissue (as collected by the authors) likely contains predominantly alveolar cell types rather than bronchial epithelial cells (as the BEAS-2B cell line). In this contact, comparing OCR in peripheral lung tissue with an alveolar cell type (for example A549) would be more logical.

  1. The authors state that the non-mitochondrial respiration between the tissues and cell lines are not statistically different. However In fig. 2C there seems to be quite a big difference in non-mitochodnrial respiration in H9C2 cardiomyocytes (+/- 30%) vs cardiac tissue (60%). Can the authors comment on this? Also, a non-mitochondrial respiration of 60% in cardiac tissue seems rather high. Did the authors try to titrate rotenone and antimycin A to show maximal inhibition of mitochondrial respiration. If so, this would be important to show. If not, this should be performed and presented in the paper (or supplement)

  1. Respirometry with Seahorse allows assessing different aspects of OCR (basal respiration, maximal respiration (FCCP), non mitochondrial, ATP-coupled respiration etc…). Can the authors show this?

Minor comments:

  1. It appears that fig S3 is missing from the document?

  1. What is the added value of Fig S1 A and B. If this indeed represents essential information than please present the graphs related to the same cell type together (so A + C next to each other and B and D)

Author Response

IJMS
A new strategy to preserve and assess oxygen consumption in murine tissues

Dear Dr Zhang,

We thank you and the reviewers sincerely for your time to read our manuscript and for formulating the following comments, suggestions, and critics. We appreciate the positive comments, and we revised our manuscript to address the extensive and insightful concerns of the reviewers.

Reviewer 1:

The study from Kluza et al is interesting and addresses an important topic i.e. measurement of tissue respiratory capacity. The authors describe a new method of assessing O2 consumption in intact tissues and addressed the viability of a ‘’storing’’ option in this methodology (storing samples overnight at 4 degrees). In addition, the authors assessed O2 consumption rate in cardiac and lung tissue of obese mice. Although the methodology and results that are being described are interesting, I do have a few concerns/remarks.

Answer: We thank the reviewer for their positive comments in our article. Following the recommendations/remarks of the reviewer, we have modified Figures 2, 3 and 4, and included new results in additional new supplemental Figures.

Statements regarding (abnormal) histology and mitochondrial ultrastructure without proper quantification of specific features warrants caution and this should be addressed. In this context, depiction of histological or ultrastructural features (e.g. Fig. 2 D, E; Fig 3D, E; Fig S2C) without further labelling of specific intracellular features (organelles) is unclear.

Answer: We agree with the referee. According to histological score of our heart and lung sections, no difference was observed between fresh samples and samples stored overnight at 4°C, in terms of inflammation, remodeling, thickness of basal membrane, structural architecture. In contrast, samples conserved overnight at 37°c are structurally totally disorganized.  We added in Figure 2 and 3 a quantification of mitochondrial size (cross-sectional area) for "fresh tissue" versus "storing samples overnight at 4°C". Approximately 100 mitochondria from 4 different hearts or lungs were analyzed. Our results show no modification of mitochondrial size in both conditions. Manuscript has been updated with this data (line 187).

Moreover, we have indicated the name of specific organelles in panel E of Figure 2 and 3 (Mitochondria, Sarcomeres, Nucleus) to clarify the interpretation of electron microscopy.

For the HFD vs control diet experiment using mice, assessment of specific blood circulatory markers (at least cardiometabolic markers such as triglycerides, cholesterol etc) including substrates for mitochondrial metabolism would strengthen the manuscript.

Answer: We agree with reviewer. We added data in Figure 4B to show the concentration of triglycerides and cholesterol in murine plasma. As expected HFD mice exhibited an increase of triglyceride and cholesterol concentration compared to control diet.

we modified sentence line 219: " As shown on Figure 4A and 4B, sixteen weeks of high-fat diet (HFD) induced a significant increase in body weight (P<0.0001) and an increase of triglyceride and cholesterol in plasma concentration when compared to low-fat diet (LFD)-fed lean animals."

In addition, OCR measurements using specific substrates (for example fatty acids e.g. palmitoyl carnithine) could shed more light on mitochondrial substrate use in the obese animals vs control animals.

The reviewer is right, the medium used to assess OCR was composed of glutamine and glucose, but no fatty acid. Previously, we tried to perform experiments using several fatty acids, but the diffusion of these substrates seems to be more difficult in non-permeabilized tissues.

The rationale and impact for assessing OCR in lung tissue in the context of obesity should be described more clear in the introduction and should be more elaborately discussed in the discussion.

Answer: We have used HFD mice to validate our protocol as proof- of-concept. In a previous Paper, Pizzola et al (7) has assessed the metabolic activity of lung single cell suspensions by measuring the oxygen consumption rate (OCR). They showed that lung cells from HFD mice exhibited higher oxygen consumption compared to cells from Normal Diet mice. Our data are in agreement with these experiments. However, the way we assess the OCR using our new technic allows the preservation of the micro-environment compared to single cell preparation and allows to take into account the multiplicity of cell types composing cardiac or lung tissue.

Assessment of HO1 mRNA as a marker for hypoxia in the tissues should be expanded to more hypoxia-related markers. For example assessment of HIF-1a protein levels or measurements of more HIF-responsive genes.

Answer: We agree with the referee. We added the expression of target genes regulated by HIF-1α: Pyruvate dehydrogenase kinase 3, lactate dehydrogenase 2 and GLUT transporter 1 SLC2A1. As seen in new supplementary Figure 5, the levels of these genes are similar in "fresh tissues" versus "storage at 4°C overnight ".

We modified line 197 to 200 by the following sentence: "Preservation of tissue overnight at 4°C did not induce hypoxia as depicted by mRNA levels of target genes regulated by HIF-1α (Enolase 3, Pyruvate dehydrogenase kinase 3, lactate dehydrogenase 2 and GLUT transporter SLC2A1, heme oxygenase 1) in contrast to 37°C in heart (Supplemental Figure S5A) and lung (Supplemental Figure S5B)"

The extent to which assessment of OCR in peripheral lung tissue vs BEAS-2B cells is a viable comparison can be debated as peripheral lung tissue (as collected by the authors) likely contains predominantly alveolar cell types rather than bronchial epithelial cells (as the BEAS-2B cell line). In this contact, comparing OCR in peripheral lung tissue with an alveolar cell type (for example A549) would be more logical.

Answer: We agree with the reviewer. We performed OCR assessment in A549 cells, and we modified Figure 3C.

The authors state that the non-mitochondrial respiration between the tissues and cell lines are not statistically different. However, in fig. 2C there seems to be quite a big difference in non-mitochondrial respiration in H9C2 cardiomyocytes (+/- 30%) vs cardiac tissue (60%). Can the authors comment on this? Also, a non-mitochondrial respiration of 60% in cardiac tissue seems rather high.

Answer: Insensitive rotenone/antimycin A respiration is a marker of oxygen-consuming processes that are not mitochondrial. Non-mitochondrial OCR is attributed to enzymes including cyclo-oxygenases, lipoxygenases and NADPH oxidases. But non-mitochondrial OCR increases also in the presence of stressor as ROS and RNS. We think that the high level of non-mitochondrial respiration could be due to stress following mice sacrifice (specially the role of ketamine/xylazine anesthesia) as shown in the paper below.

Venâncio C et al., Acute ketamine impairs mitochondrial function and promotes superoxide dismutase activity in the rat brain. Anesth Analg. 2015 Feb;120(2):320-8. doi: 10.1213/ANE.0000000000000539;

Kevin LG et al.  Reactive oxygen species as mediators of cardiac injury and protection: the relevance to anesthesia practice. Anesth Analg. 2005 Nov;101(5):1275-1287. doi: 10.1213/01.ANE.0000180999.81013.D0.)

The following sentence is now added in the manuscript: " Nonetheless, we observed elevated non-mitochondrial respiration in cardiac tissue which may be due to stress following anesthesia prior to mouse sacrifice (Venancio et al, 2015).)

Did the authors try to titrate rotenone and antimycin A to show maximal inhibition of mitochondrial respiration. If so, this would be important to show. If not, this should be performed and presented in the paper (or supplement)

Answer: As requested by the reviewer, titrations of rotenone and antimycin A have been realized and confirmed that concentrations (10µM) used in this study induced a full inhibition of mitochondrial respiration. These results are now included in supplementary Figure S1 and we added the following sentence in the manuscript (line 122-124): "First, we realized a titration of rotenone and antimycin A to determine the appropriate concentration to reach the maximal inhibition of mitochondrial respiration (see new supplementary Figure S1)."

Respirometry with Seahorse allows assessing different aspects of OCR (basal respiration, maximal respiration (FCCP), nonmitochondrial, ATP-coupled respiration etc…). Can the authors show this?

Answer: To assess the different aspects of OCR, we need to determine the incubation time and concentration needed for specific inhibitors (FCCP, oligomycin A, etc ...) for each tissue type. This work is under progress but requires a longer investigation; it will be the subject of a future article.

 It appears that fig S3 is missing from the document?

Answer: We apologize for this mistake, the figured is now added to the revised manuscript.

What is the added value of Fig S1 A and B. If this indeed represents essential information than please present the graphs related to the same cell type together (so A + C next to each other and B and D)

Answer: We think that results in Figure S1 could be interesting for readers; specially to compare OCR profile of cell lines versus tissue. But to simplify the figure, we have removed panel A and B including the oxygen quantification in mm Hg.

Reviewers 2

The authors explored the mitochondrial respiration on heart and lung slices using Seahorse. First, the authors evaluate mitochondrial respiration at different timing and they observed that temperature is a critical factor to maintain muscle structure and respiratory function. In a second experimental protocol they compared respiratory function of obese and WT mice and they showed that obesity is associated with respiratory alteration. Overall, it is difficult to understand what was the main objective of this study

Answer: We thank the reviewer for his/her comments in our paper. We hope that our changes (see below) have clarified the objectives of our study.

The main limiting factor is that no gold standard measurements are used. The authors explored respiration in H9C2 and BEAS-2B cells (supplemental figure 1) but it is difficult to compared respiration as the units are different (pmol/min/ug protein vs. pmol/min). Was the ratio similar between respiration with or without rotenone and Antimycin A?

Answer: Rotenone and antimycin A are mainly used for inhibition of mitochondrial respiration for assessment of OXPHOS in monolayer cell lines (as H9C2, Beas-2B), cardiac permeabilized fibers or isolated mitochondria. But theses inhibitors are more rarely used in non-permeabilized tissue. We used cells lines to validate the efficiency of the mixture before to use it in non-permeabilized lung or cardiac tissue. Of course, our objective is not to compared metabolic organization between tissue and cell lines.

We have modified the manuscript to make it clearer. "Rotenone and Antimycin A inhibited mitochondrial oxygen consumption in both models, but there is a more important insensitive rotenone/antimycin respiration in the tissues than in corresponding the cell lines, in particular for the heart (Figures 2C and 3C, Supplemental Figure S2). Similar results have been obtained with potassium cyanide (KCN), an inhibitor of the complex IV of the mitochondrial respiratory chain (Supplemental Figure S3A and S3B)."

To note, Beas2be cells have been replaced by A549 cell lines as requested by reviewer 1.

The authors should emphasis on the practical interest of this new technique. What is the different with traditional mitochondrial assessment (isolated mitochondrial, permeabilized fibers)?

Answer: Our goal is to provide a complementary method for the analysis of mitochondrial metabolism in non-permeabilized mouse tissue. Our introduction (l52-60) addressed the limits of traditional techniques:

"Ex vivo techniques on isolated cells and mitochondria may not adequately reflect the in vivo status since cells and organelles will not be in their tissular and cellular micro-environments, respectively. Moreover, isolation procedures often altered cell and organelle structures and functions, notably through disrupting the mitochondrial network, due to the loss of intracellular interactions. Most often individual cell populations are analyzed, losing the cell-to-cell interactions which occur in the whole tissue. Analysis of permeabilized cells induced a lack of most of the cytoplasm (thus the impact of cytosolic factors and intra-tissular nutrients cannot be studied) and a risk of partial or excessive permeabilization. "

 From line 71 to 74, we added now the sentence to clarify our objectives: In the present study, we proposed a complementary approach to assess mitochondrial metabolism in non-permeabilized lung and heart tissue.

The authors point out the interest in the context of transplantation. Is respiratory function a criterion that would lead to organ dysfunction after surgery?

Answer: In recent paper published (2021) in trends in Molecular Medicine, Saeb-Parsy et al have underlined how the metabolic changes that occur within the organ during transplantation, particularly those associated with mitochondria, may contribute to the outcome. This reference is now added in our discussion.

Moreover, in the context of transplantation, the assessment using traditional mitochondrial respiration technique will require the same time and will allow to explore maximal mitochondrial capacities.

Of course, the traditional respiration technique (permeabilized tissue, isolated mitochondria) will be also efficient to assess oxygen consumption in tissue prior to transplantation. In this study, we propose for the first time an alternative method using XF Seahorse in non-permeabilized tissue. To our knowledge, this had never been evaluated.

It would be interesting to provide the coefficient of variation of the measure. For example, for LDH measurement in cardiac muscle, there is a visual ratio between the samples of 4 after a conservation at 4°C overnight.

Answer: As observed by the reviewer, variations are observed. These could probably due to in vivo heterogeneity.

As LDH was used as a marker of muscle damage, it seems therefore intriguing that there is no difference of mitochondrial respiration or HO1 mRNA expression between fresh samples and 4°C overnight samples.

Answer: To confirm this result (and as requested by reviewer 1), we added the expression of target genes regulated by HIF-1α: Pyruvate dehydrogenase kinase 3, lactate dehydrogenase 2 and GLUT transporter SLC2A1. As seen in new supplementary figure 5, the levels of these genes are similar in "fresh tissues" versus "storage at 4°C overnight ".

Moreover, the number of animals n the second study is important and some values of obeses group are quite similar to control. It questions the sensibility of this technique.

Answer: As observed by the reviewer, variations are observed despite the high number of samples (at least 20 mice). These could probably be explained by in vivo heterogeneity.

I would suggest to reduce introduction length to focus on the main objective of the study.

Answer: We agree, and we deleted line 48 to 51 and line 60-64.

L56-57. It would be interesting to add the reference of Picard et al. 2011 (PMID: 21512578).

Answer: As requested, this reference is now added in the manuscript.

L66-68: NIRS allow to measure tissue O2 saturation, changes in hemoglobin volume. While it reflects muscle oxygenation, it doesn’t reflect properly neither oxygen consumption nor mitochondrial respiration as the blood flow is unknown.

Answer: We agree with the reviewer, and we have deleted this sentence.

L86-106. It is unclear to me how the surface area was measured. The information are inconsistent between the text and the figure 1. The diameter should be expressed in mm and not in mm2.

Answer: Tissue area were measure with countuor tools in Zen 2.3 software. It is an area measurement so unity is well mm2. We replace the term “size” by “area” in the main text to avoid confusion.

Figure 4: the number of samples was different from that claimed in the methods section.

Answer: The number of mice has been changed the legend of Figure 4.

What is the rational about the measurement after 24h? Whenever several animals are sacrificed in a same day, we will not wait the next day to start

Answer: We think that scientists involved in the energy metabolism field could be interesting to know that storage at 4 ° C (limited to one night) will not drastically modify the mitochondrial metabolism of the tissues. As written in the discussion section, "preserving samples overnight at 4°C could be useful to collect many samples from large cohorts and to compare their OCR on the same plate, limiting experimental errors"

Reviewer 2 Report

The authors explored the mitochondrial respiration on heart and lung slices using Seahorse. First, the authors evaluate mitochondrial respiration at different timing and they observed that temperature is a critical factor to maintain muscle structure and respiratory function. In a second experimental protocol they compared respiratory function of obese and WT mice and they showed that obesity is associated with respiratory alteration. Overall, it is difficult to understand what was the main objective of this study

I point out some major limitations that should be considered or at least discussed:

  • The main limiting factor is that no gold standard measurements are used. The authors explored respiration in H9C2 and BEAS-2B cells (supplemental figure 1) but it is difficult to compared respiration as the units are different (pmol/min/ug protein vs. pmol/min). Was the ratio similar between respiration with or without rotenone and Antimycin A?
  • The authors should emphasis on the practical interest of this new technique. What is the different with traditional mitochondrial assessment (isolated mitochondrial, permeabilized fibers)? The authors point out the interest in the context of transplantation. Is respiratory function a criterion that would lead to organ dysfunction after surgery? Moreover, in the context of transplantation, the assessment using traditional mitochondrial respiration technique will require the same time and will allow to explore maximal mitochondrial capacities.
  • It would be interesting to provide the coefficient of variation of the measure. For example for LDH measurement in cardiac muscle, there is a visual ratio between the samples of 4 after a conservation at 4°C overnight. As LDH was used as a marker of muscle damage, it seems therefore intriguing that there is no difference of mitochondrial respiration or HO1 mRNA expression between fresh samples and 4°C overnight samples. Moreover, the number of animals n the second study is important and some values of obeses group are quite similar to control. It questions the sensibility of this technique.

Minor comments:

  • I would suggest to reduce introduction length to focus on the main objective of the study.
  • 56-57. It would be interesting to add the reference of Picard et al. 2011 (PMID: 21512578).
  • L66-68: NIRS allow to measure tissue O2 saturation, changes in hemoglobin volume. While it reflects muscle oxygenation, it doesn’t reflect properly neither oxygen consumption nor mitochondrial respiration as the blood flow is unknown.
  • L86-106. It is unclear to me how the surface area was measured. The information are inconsistent between the text and the figure 1. The diameter should be expressed in mm and not in mm2.
  • Figure 4: the number of samples was different from that claimed in the methods section.
  • What is the rational about the measurement after 24h? Whenever several animals are sacrificed in a same day, we will not wait the next day to start the experiment?

Author Response

IJMS
A new strategy to preserve and assess oxygen consumption in murine tissues

Dear Dr Zhang,

We thank you and the reviewers sincerely for your time to read our manuscript and for formulating the following comments, suggestions, and critics. We appreciate the positive comments, and we revised our manuscript to address the extensive and insightful concerns of the reviewers.

Reviewer 1:

The study from Kluza et al is interesting and addresses an important topic i.e. measurement of tissue respiratory capacity. The authors describe a new method of assessing O2 consumption in intact tissues and addressed the viability of a ‘’storing’’ option in this methodology (storing samples overnight at 4 degrees). In addition, the authors assessed O2 consumption rate in cardiac and lung tissue of obese mice. Although the methodology and results that are being described are interesting, I do have a few concerns/remarks.

Answer: We thank the reviewer for their positive comments in our article. Following the recommendations/remarks of the reviewer, we have modified Figures 2, 3 and 4, and included new results in additional new supplemental Figures.

Statements regarding (abnormal) histology and mitochondrial ultrastructure without proper quantification of specific features warrants caution and this should be addressed. In this context, depiction of histological or ultrastructural features (e.g. Fig. 2 D, E; Fig 3D, E; Fig S2C) without further labelling of specific intracellular features (organelles) is unclear.

Answer: We agree with the referee. According to histological score of our heart and lung sections, no difference was observed between fresh samples and samples stored overnight at 4°C, in terms of inflammation, remodeling, thickness of basal membrane, structural architecture. In contrast, samples conserved overnight at 37°c are structurally totally disorganized.  We added in Figure 2 and 3 a quantification of mitochondrial size (cross-sectional area) for "fresh tissue" versus "storing samples overnight at 4°C". Approximately 100 mitochondria from 4 different hearts or lungs were analyzed. Our results show no modification of mitochondrial size in both conditions. Manuscript has been updated with this data (line 187).

Moreover, we have indicated the name of specific organelles in panel E of Figure 2 and 3 (Mitochondria, Sarcomeres, Nucleus) to clarify the interpretation of electron microscopy.

For the HFD vs control diet experiment using mice, assessment of specific blood circulatory markers (at least cardiometabolic markers such as triglycerides, cholesterol etc) including substrates for mitochondrial metabolism would strengthen the manuscript.

Answer: We agree with reviewer. We added data in Figure 4B to show the concentration of triglycerides and cholesterol in murine plasma. As expected HFD mice exhibited an increase of triglyceride and cholesterol concentration compared to control diet.

we modified sentence line 219: " As shown on Figure 4A and 4B, sixteen weeks of high-fat diet (HFD) induced a significant increase in body weight (P<0.0001) and an increase of triglyceride and cholesterol in plasma concentration when compared to low-fat diet (LFD)-fed lean animals."

In addition, OCR measurements using specific substrates (for example fatty acids e.g. palmitoyl carnithine) could shed more light on mitochondrial substrate use in the obese animals vs control animals.

The reviewer is right, the medium used to assess OCR was composed of glutamine and glucose, but no fatty acid. Previously, we tried to perform experiments using several fatty acids, but the diffusion of these substrates seems to be more difficult in non-permeabilized tissues.

The rationale and impact for assessing OCR in lung tissue in the context of obesity should be described more clear in the introduction and should be more elaborately discussed in the discussion.

Answer: We have used HFD mice to validate our protocol as proof- of-concept. In a previous Paper, Pizzola et al (7) has assessed the metabolic activity of lung single cell suspensions by measuring the oxygen consumption rate (OCR). They showed that lung cells from HFD mice exhibited higher oxygen consumption compared to cells from Normal Diet mice. Our data are in agreement with these experiments. However, the way we assess the OCR using our new technic allows the preservation of the micro-environment compared to single cell preparation and allows to take into account the multiplicity of cell types composing cardiac or lung tissue.

Assessment of HO1 mRNA as a marker for hypoxia in the tissues should be expanded to more hypoxia-related markers. For example assessment of HIF-1a protein levels or measurements of more HIF-responsive genes.

Answer: We agree with the referee. We added the expression of target genes regulated by HIF-1α: Pyruvate dehydrogenase kinase 3, lactate dehydrogenase 2 and GLUT transporter 1 SLC2A1. As seen in new supplementary Figure 5, the levels of these genes are similar in "fresh tissues" versus "storage at 4°C overnight ".

We modified line 197 to 200 by the following sentence: "Preservation of tissue overnight at 4°C did not induce hypoxia as depicted by mRNA levels of target genes regulated by HIF-1α (Enolase 3, Pyruvate dehydrogenase kinase 3, lactate dehydrogenase 2 and GLUT transporter SLC2A1, heme oxygenase 1) in contrast to 37°C in heart (Supplemental Figure S5A) and lung (Supplemental Figure S5B)"

The extent to which assessment of OCR in peripheral lung tissue vs BEAS-2B cells is a viable comparison can be debated as peripheral lung tissue (as collected by the authors) likely contains predominantly alveolar cell types rather than bronchial epithelial cells (as the BEAS-2B cell line). In this contact, comparing OCR in peripheral lung tissue with an alveolar cell type (for example A549) would be more logical.

Answer: We agree with the reviewer. We performed OCR assessment in A549 cells, and we modified Figure 3C.

The authors state that the non-mitochondrial respiration between the tissues and cell lines are not statistically different. However, in fig. 2C there seems to be quite a big difference in non-mitochondrial respiration in H9C2 cardiomyocytes (+/- 30%) vs cardiac tissue (60%). Can the authors comment on this? Also, a non-mitochondrial respiration of 60% in cardiac tissue seems rather high.

Answer: Insensitive rotenone/antimycin A respiration is a marker of oxygen-consuming processes that are not mitochondrial. Non-mitochondrial OCR is attributed to enzymes including cyclo-oxygenases, lipoxygenases and NADPH oxidases. But non-mitochondrial OCR increases also in the presence of stressor as ROS and RNS. We think that the high level of non-mitochondrial respiration could be due to stress following mice sacrifice (specially the role of ketamine/xylazine anesthesia) as shown in the paper below.

Venâncio C et al., Acute ketamine impairs mitochondrial function and promotes superoxide dismutase activity in the rat brain. Anesth Analg. 2015 Feb;120(2):320-8. doi: 10.1213/ANE.0000000000000539;

Kevin LG et al.  Reactive oxygen species as mediators of cardiac injury and protection: the relevance to anesthesia practice. Anesth Analg. 2005 Nov;101(5):1275-1287. doi: 10.1213/01.ANE.0000180999.81013.D0.)

The following sentence is now added in the manuscript: " Nonetheless, we observed elevated non-mitochondrial respiration in cardiac tissue which may be due to stress following anesthesia prior to mouse sacrifice (Venancio et al, 2015).)

Did the authors try to titrate rotenone and antimycin A to show maximal inhibition of mitochondrial respiration. If so, this would be important to show. If not, this should be performed and presented in the paper (or supplement)

Answer: As requested by the reviewer, titrations of rotenone and antimycin A have been realized and confirmed that concentrations (10µM) used in this study induced a full inhibition of mitochondrial respiration. These results are now included in supplementary Figure S1 and we added the following sentence in the manuscript (line 122-124): "First, we realized a titration of rotenone and antimycin A to determine the appropriate concentration to reach the maximal inhibition of mitochondrial respiration (see new supplementary Figure S1)."

Respirometry with Seahorse allows assessing different aspects of OCR (basal respiration, maximal respiration (FCCP), nonmitochondrial, ATP-coupled respiration etc…). Can the authors show this?

Answer: To assess the different aspects of OCR, we need to determine the incubation time and concentration needed for specific inhibitors (FCCP, oligomycin A, etc ...) for each tissue type. This work is under progress but requires a longer investigation; it will be the subject of a future article.

 It appears that fig S3 is missing from the document?

Answer: We apologize for this mistake, the figured is now added to the revised manuscript.

What is the added value of Fig S1 A and B. If this indeed represents essential information than please present the graphs related to the same cell type together (so A + C next to each other and B and D)

Answer: We think that results in Figure S1 could be interesting for readers; specially to compare OCR profile of cell lines versus tissue. But to simplify the figure, we have removed panel A and B including the oxygen quantification in mm Hg.

Reviewer 2

The authors explored the mitochondrial respiration on heart and lung slices using Seahorse. First, the authors evaluate mitochondrial respiration at different timing and they observed that temperature is a critical factor to maintain muscle structure and respiratory function. In a second experimental protocol they compared respiratory function of obese and WT mice and they showed that obesity is associated with respiratory alteration. Overall, it is difficult to understand what was the main objective of this study

Answer: We thank the reviewer for his/her comments in our paper. We hope that our changes (see below) have clarified the objectives of our study.

The main limiting factor is that no gold standard measurements are used. The authors explored respiration in H9C2 and BEAS-2B cells (supplemental figure 1) but it is difficult to compared respiration as the units are different (pmol/min/ug protein vs. pmol/min). Was the ratio similar between respiration with or without rotenone and Antimycin A?

Answer: Rotenone and antimycin A are mainly used for inhibition of mitochondrial respiration for assessment of OXPHOS in monolayer cell lines (as H9C2, Beas-2B), cardiac permeabilized fibers or isolated mitochondria. But theses inhibitors are more rarely used in non-permeabilized tissue. We used cells lines to validate the efficiency of the mixture before to use it in non-permeabilized lung or cardiac tissue. Of course, our objective is not to compared metabolic organization between tissue and cell lines.

We have modified the manuscript to make it clearer. "Rotenone and Antimycin A inhibited mitochondrial oxygen consumption in both models, but there is a more important insensitive rotenone/antimycin respiration in the tissues than in corresponding the cell lines, in particular for the heart (Figures 2C and 3C, Supplemental Figure S2). Similar results have been obtained with potassium cyanide (KCN), an inhibitor of the complex IV of the mitochondrial respiratory chain (Supplemental Figure S3A and S3B)."

To note, Beas2be cells have been replaced by A549 cell lines as requested by reviewer 1.

The authors should emphasis on the practical interest of this new technique. What is the different with traditional mitochondrial assessment (isolated mitochondrial, permeabilized fibers)?

Answer: Our goal is to provide a complementary method for the analysis of mitochondrial metabolism in non-permeabilized mouse tissue. Our introduction (l52-60) addressed the limits of traditional techniques:

"Ex vivo techniques on isolated cells and mitochondria may not adequately reflect the in vivo status since cells and organelles will not be in their tissular and cellular micro-environments, respectively. Moreover, isolation procedures often altered cell and organelle structures and functions, notably through disrupting the mitochondrial network, due to the loss of intracellular interactions. Most often individual cell populations are analyzed, losing the cell-to-cell interactions which occur in the whole tissue. Analysis of permeabilized cells induced a lack of most of the cytoplasm (thus the impact of cytosolic factors and intra-tissular nutrients cannot be studied) and a risk of partial or excessive permeabilization. "

From line 71 to 74, we added now the sentence to clarify our objectives: In the present study, we proposed a complementary approach to assess mitochondrial metabolism in non-permeabilized lung and heart tissue.

The authors point out the interest in the context of transplantation. Is respiratory function a criterion that would lead to organ dysfunction after surgery?

Answer: In recent paper published (2021) in trends in Molecular Medicine, Saeb-Parsy et al have underlined how the metabolic changes that occur within the organ during transplantation, particularly those associated with mitochondria, may contribute to the outcome. This reference is now added in our discussion.

Moreover, in the context of transplantation, the assessment using traditional mitochondrial respiration technique will require the same time and will allow to explore maximal mitochondrial capacities.

Of course, the traditional respiration technique (permeabilized tissue, isolated mitochondria) will be also efficient to assess oxygen consumption in tissue prior to transplantation. In this study, we propose for the first time an alternative method using XF Seahorse in non-permeabilized tissue. To our knowledge, this had never been evaluated.

It would be interesting to provide the coefficient of variation of the measure. For example, for LDH measurement in cardiac muscle, there is a visual ratio between the samples of 4 after a conservation at 4°C overnight.

Answer: As observed by the reviewer, variations are observed. These could probably due to in vivo heterogeneity.

As LDH was used as a marker of muscle damage, it seems therefore intriguing that there is no difference of mitochondrial respiration or HO1 mRNA expression between fresh samples and 4°C overnight samples.

Answer: To confirm this result (and as requested by reviewer 1), we added the expression of target genes regulated by HIF-1α: Pyruvate dehydrogenase kinase 3, lactate dehydrogenase 2 and GLUT transporter SLC2A1. As seen in new supplementary figure 5, the levels of these genes are similar in "fresh tissues" versus "storage at 4°C overnight ".

Moreover, the number of animals n the second study is important and some values of obeses group are quite similar to control. It questions the sensibility of this technique.

Answer: As observed by the reviewer, variations are observed despite the high number of samples (at least 20 mice). These could probably be explained by in vivo heterogeneity.

I would suggest to reduce introduction length to focus on the main objective of the study.

Answer: We agree, and we deleted line 48 to 51 and line 60-64.

L56-57. It would be interesting to add the reference of Picard et al. 2011 (PMID: 21512578).

Answer: As requested, this reference is now added in the manuscript.

L66-68: NIRS allow to measure tissue O2 saturation, changes in hemoglobin volume. While it reflects muscle oxygenation, it doesn’t reflect properly neither oxygen consumption nor mitochondrial respiration as the blood flow is unknown.

Answer: We agree with the reviewer, and we have deleted this sentence.

L86-106. It is unclear to me how the surface area was measured. The information are inconsistent between the text and the figure 1. The diameter should be expressed in mm and not in mm2.

Answer: Tissue area were measure with countuor tools in Zen 2.3 software. It is an area measurement so unity is well mm2. We replace the term “size” by “area” in the main text to avoid confusion.

Figure 4: the number of samples was different from that claimed in the methods section.

Answer: The number of mice has been changed the legend of Figure 4.

What is the rational about the measurement after 24h? Whenever several animals are sacrificed in a same day, we will not wait the next day to start

Answer: We think that scientists involved in the energy metabolism field could be interesting to know that storage at 4 ° C (limited to one night) will not drastically modify the mitochondrial metabolism of the tissues. As written in the discussion section, "preserving samples overnight at 4°C could be useful to collect many samples from large cohorts and to compare their OCR on the same plate, limiting experimental errors"

Round 2

Reviewer 1 Report

The authors have addressed all my concerns and questions